

# Proteomic analysis of IgM antigens from mammary tissue under pre- and post-cancer conditions using the MMTV-PyVT mouse model

Ricardo Hernández Ávila[1], Mariana Díaz-Zaragoza[2] and Pedro Ostoa-Saloma[1]

[1] Departamento de Inmunología, Instituto de Investigaciones Biomédicas, Universidad Nacional Autonoma de México, Ciudad de México, CdMx, México
[2] Laboratorio de Sistemas Biológicos, Departamento de Ciencias de la Salud. Centro Universitario de los Valles, Universidad de Guadalajara, Ameca, Jalisco, México

## ABSTRACT

We analyzed the recognition of tumor antigens by IgM in transgenic MMTV-PyVT mice. PyVT female mice are a model of breast cancer that simulates its counterpart in humans. The PyVT model allows studying antigen recognition in two conditions: before and during tumor expression. We attempted to identify by sequence, the antigens recognized by IgM that are expressed or disappear in the membrane of breast transgenic tissue during the transition "No tumor-Tumor". 2D immunoblots were obtained of isolated membranes from the breast tissue in the fifth, sixth, and seventh week (transition point). Proteins recognized by IgM were sequenced in duplicate by MALDI-TOF. In the transition, we observed the disappearance of antigens in transgenic mice with respect to non-transgenic ones. We believe that in the diagnosis of cancer in its early stages, the expression of early antigens is as important as their early delocalization, with the latter having the advantage that, under normal conditions, we can know which proteins should be present at a given time. Therefore, we could consider that also the absence of antigens could be considered as a biomarker of cancer in progress.

## INTRODUCTION

Tumors express aberrant, mutated, or modified proteins that are associated with malignant growth. These proteins are known as tumor-associated antigens and the carbohydrates in these proteins are tumor-associated carbohydrate antigens. These antigens can stimulate the humoral and cellular immune responses (*Disis et al., 1997*; *Carter, Smith & Ryan, 2004*). Carbohydrate epitopes are stably expressed in many tumors at various precursor stages. Unlike polypeptide chains, glycoepitopes share structural homologies beyond the boundaries of protein families; therefore, they can cross-react and be prime targets for natural IgM antibodies (*Vollmers & Brändlein, 2006*).

Two types of IgM can be found in the blood of normal mice: Natural IgM, secreted by CD5 B-1 cells in the absence of antigenic stimulation and which concentrates most of the circulating IgM. On the other hand, antigen-induced IgM is produced primarily by B cells

Corresponding author
Pedro Ostoa-Saloma,
postoa@unam.mx

(B-2) only after antigenic stimulation. Both natural and induced IgM are polymeric and efficiently activate the classical complement cascade. Natural antibodies exhibit a broad spectrum of antibacterial activity and serve as the first line of defense against microbial and viral infections (*Baumgarth et al., 2000*; *Klimovich, 2011*). Also, IgM is associated with the recognition and elimination of precancerous and cancerous cells (*Vollmers & Brändlein, 2009*; *Vollmers & Brändlein, 2005*; *Díaz-Zaragoza et al., 2015b*).

Previously, we demonstrated the advantage of IgM with respect to IgG in the recognition of tumor antigens (*Díaz-Zaragoza et al., 2015a*; *Ostoa-Saloma et al., 2009*). In that article, variability in the IgM response was manifested as a pattern of spots could represent a signature of the immune response. Different numbers of spots was found in the IgG and IgM responses. On average, the IgM had more and stronger response than IgG but also the latter decrease with the time. In this context our question was: In an organism that we know will have breast cancer, will its pattern of antigenic recognition by immunoglobulin M vary at the time of transition?

For several years, two-dimensional polyacrylamide gel electrophoresis (2D-PAGE), followed by mass spectrometry for identifying proteins, has been the main technique for the discovery of biomarkers in conventional proteomic analysis (*Gorg et al., 2000*; *Hanash, 2000*). This technique is especially suitable for making direct comparisons of the differential expression of proteins between normal and tumor tissues (*Soldes et al., 1999*; *Seow et al., 2001*; *Celis et al., 1999*; *Celis, Wolf & Ostergaard, 2000*; *Celis et al., 2002*; *Chen et al., 2002*; *Franzen et al., 1997*). Recently, Nuclear magnetic resonance (NMR)-based metabolomics to a small blood sample offers a complementary approach to the current pathway for investigating patients with a clinical suspected possible malignancy. It is sensitive, specific, and of low cost, requiring nothing more than a blood sample in the clinic and an inexpensive NMR analysis, and can identify patients with solid tumors when referred with nonspecific symptoms (*Larkin et al., 2022*).

Mouse Mammary Tumor Virus promoter-Polyoma Virus Middle T Antigen (MMTV-PyVT) mice is a murine model that has been genetically modified to develop breast cancer with characteristics similar to human breast cancer. We believe that, with this model, we can achieve our goal of comparing the pattern of antigen recognition by the IgM in two conditions: before and during the expression of the tumor. In the present study, we report the pattern of antigenic recognition by IgM in serum of transgenic or non-transgenic mice with the membrane fraction of breast tissue at the sixth week, which is the starting period of the transformation process. Therefore, we sequenced the antigens recognized at that time shortly after the Middle T Antigen (MT) antigen gene was expressed.

## MATERIALS & METHODS

### Ethics statement

Animal care and experimental practice were conducted at the Unidad de Modelos Biológicos (UMB) in the Instituto de Investigaciones Biomédicas (IIB), Universidad Nacional Autónoma de México. All experimental procedures in the animals were approved by the Institutional Care and Animal Use Committee (CICUAL), permit number ID

264, adhering to Mexican regulation (NOM-062-ZOO-1999), and in accordance with the recommendations from the National Institute of Health (NIH) of the United States of America (Guide for the Care and Use of Laboratory Animals), available online at: https://www.biomedicas.unam.mx/servicios/unidad-de-modelos-biologicos. Euthanasia of experimental animals and controls was performed humanely by cervical dislocation after anesthesia with 5% sevofluorane (Abbot, Mexico City, México).

## Animals

Transgenic male FVB/N-Tg (MMTV-PyVT) 634Mul/J mice and FVB/N female mice were purchased from Jackson Laboratories (USA). The animals were housed at UMB at a controlled temperature (22 °C) and 12-h light-dark cycles, fed with water and Purina LabDiet 5015 (Purina, St. Louis, MO, USA) chow ad libitum, bred in the animal facilities at the UMB, and placed in standard cages with 5 mice per cage. The experimental design considers one independent variable: the antigen recognition pattern during mammary tumor induction (non-transgenic control *vs* transgenic group). Therefore statistical analysis is not applicable. Four transgenic female mice and four non transgenic female mice were used. The animals in the different experimental groups were treated and assessed at the same time. No randomisation was used to allocate experimental units to no transgenic and transgenic groups. The study did not have humane endpoints.

## DNA extraction and PCR

A 0.5 cm segment of tail tissue was cut from the 3-week-old mice. The tissue was processed according to the instructions of the DNA Wizard® Genomic DNA Purification Kit (Promega, Madison, WI, USA). Transgenic mice were recognized by PCR using the following primers for PyVT oncogene: 5′GGGAAGCAAGTACTTCACAAGG 3′ (forward) and 5′GGAAAGTCACTAGGAGCAGGG 3′ (reverse). As load control, $\beta$ actin primers 5′GGAGGTCATCACTATTGGCAACGAG 3′ (forward) and 5′TGTTTACGGATGTCAACGTCACACT 3′ (reverse) were employed.

## Serum

Sera from individual mice were obtained at the fifth, sixth, and seventh week. We incubated the blood at 4 °C for 30 min and then centrifuged it to obtain the serum. The sera were stored at −20 °C until used.

## Isolation of the membrane fraction of the breast tissue

Breast tissue at fifth, sixth, and seventh week was homogenized in PBS pH 7.4 in the presence of a protease inhibitor. The homogenate was centrifuged at 14 000 G for 15 min. The supernatant was transferred to a polycarbonate tube and centrifuged in a 90Ti rotor (Beckman, Brea, CA, USA) at 100,000 G for 1 h at 4 °C. The pellet was resuspended in lysis buffer for 2D Electrophoresis: 6 M urea, 50 mMDTT, 2% CHAPS in the presence of a protease inhibitors cocktail (Thermo Scientific, Waltham, MA, USA).

## 2D electrophoresis

For this part, the methodology described in *Díaz-Zaragoza et al. (2015a)* was followed.

## Mass spectrometry and protein identification

The samples were prepared for mass spectrometry analysis with a slight modification of a previously described procedure (*Díaz-Zaragoza et al., 2020*). The data processing software used was Protein Lynxs Global Server 2.5.1 (TM Waters). The base of Mus musculus, extracted from UNIPROT, was used for the identification of proteins.

## RESULTS

Figure 1 shows a representative amplification of MT oncogene in the newborn positive female mice selected for subsequent experiments. During the first six weeks, the mice were active and normal in appearance. As the 9th week approached, mobility decreased.

To identify antigens recognized by IgM shortly after the MT gene was expressed (sixth week), the membrane proteins of both normal and transgenic mammary tissue were separated in the fifth, sixth, and seventh weeks (there were only differences in the recognition pattern in the sixth week; therefore, the analysis was focused on this time). At the sixth week, there was variability in the antigen recognition pattern; each female presents its own antigenic pattern. However, in all cases it was observed that some antigens were recognized in the non-transgenic female but not in the transgenic female. For example, in Fig. 2, four conspicuous antigens can be observed, which were recognized by the IgM of the normal female and were not recognized by the transgenic female. These four antigens caught our attention and were sequenced in duplicate for identification.

Table 1 shows the identified proteins that coincided in the sequencing of two independent samples. Spots A, B, C, and D were sequenced in duplicate and the identification results are presented. The sequencing of spot A resulted in a protein that is encoded by the 2210010C04Rik gene and that corresponds to a Peptidase S1 domain-containing protein. The sequencing of spot B yielded two consensus proteins in the two independent analyses: a Peptidase S1 domain-containing protein and Keratin 78. Spot C also yielded two consensus proteins: Keratin type II cytoskeletal 73, and Keratin type I cytoskeletal 10. Spot D gave one consensus protein: Keratin type II cytoskeletal 6B. At the sixth week, normal breast tissue expresses keratins related to the cytoskeleton and proteases in its membrane.

## DISCUSSION

The transgenic mice FVB/N-Tg (MMTV-PyVT) 634Mul/J (transgenic mice PyVT) express the oncogene of Middle T Antigen (MT), which induces the transformation of cells from mammary tissue to a tumorigenic state; thus, the mice have a predisposition to developing breast cancer (*Guy, Cardiff & Muller, 1992*). *Zagozdzon et al. (2012)* showed that, in transgenic PyVT mice, the expression of the MT gene begins at the age of 5 weeks, thus initiating the tumor process. The membranal fraction of the mammary tissue at the fifth week did not show differences in antigenic identification by IgM (data not shown), but at the sixth week, it did. The results on the recognition of antigens distinguished by IgM one week after the MT gene was expressed can be explained because the MT antigen, when expressed, inserts into the membrane due to a contiguous 22 amino acid hydrophobic segment near the carboxyl terminal. The MT antigen has been shown to interact with

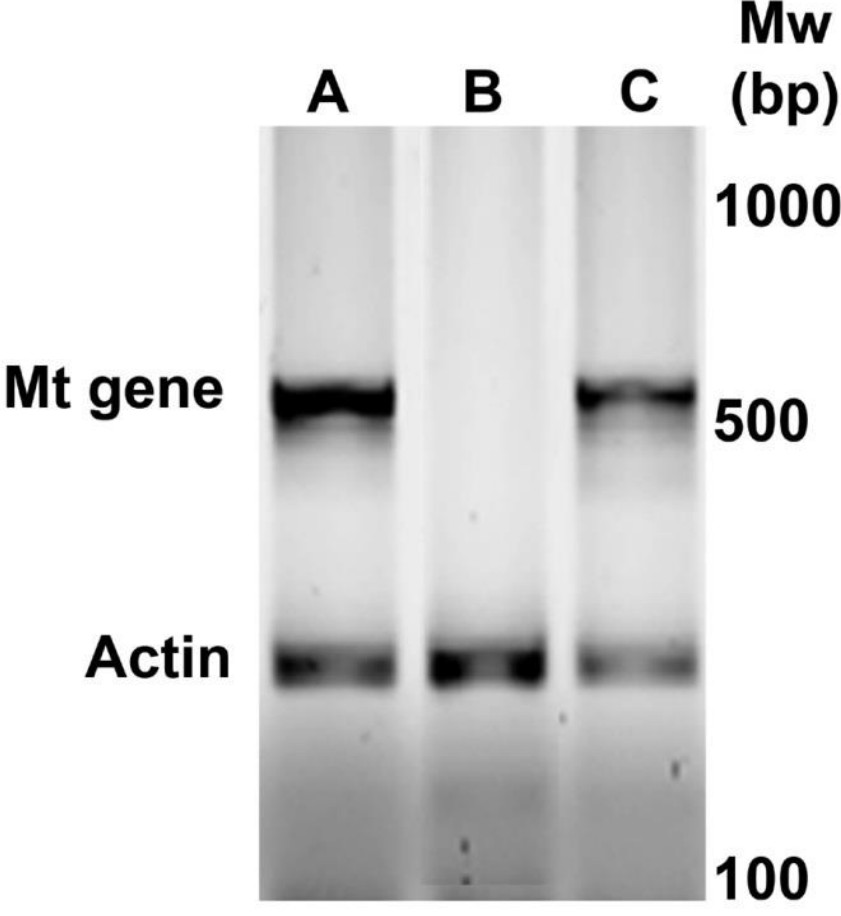

**Figure 1** **Identification by PCR of newborn female PyVT mice carrying the MT oncogene. The gel shows representative results for positive female mice (A and C) and negative female mice (B) for MT oncogene amplification.**

members of the Src family of tyrosine kinases, phosphoinositol-3-kinase (PI-3-kinase), and phosphatase 2A, altering their activity and localization. These proteins, in turn, play a role in cytoskeletal regulation, since the expression of the MT antigen generates profound changes in actin stress fibers, focal adhesions, and the regulation of microtubule arrangement. Our results confirm that the interaction of MT with membranes may involve components of the cytoskeleton. The non-detection of membrane proteins by the IgM of transgenic females is therefore explained by the alterations in cell morphology that accompany the transformation (*Andrews, Gupta & Abisdris, 1993*; *Da Costa et al., 2000*; *Horníková, Bruštíková & Forstová, 2020*; *Zheng & Foster, 2009*).

In the genesis of normal breast tissue, there are structural, compositional, and functional changes during an individual's life. The mammary epithelium undergoes dramatic growth, differentiation, and function responses to hormonal stimuli during the four stages of the mammary development cycle represented in virgin, pregnant, lactating, and involutional animals. Primary glandular branching morphogenesis occurs at approximately five weeks in

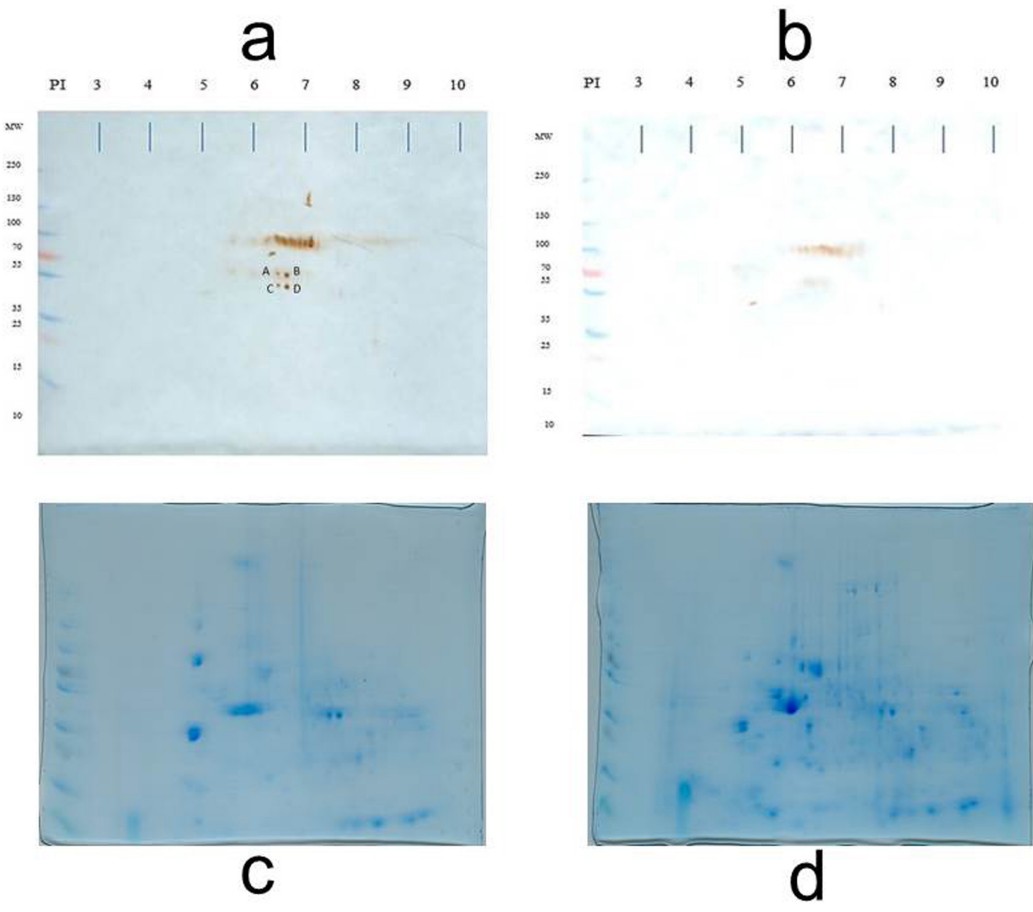

**Figure 2** **Loading control.** Representative 2D immunoblot of antigenic recognition by serum IgM of normal female mice (A) and transgenic mice (B) on the membrane fraction of mammary tissue at the sixth week of age. Blots C and D represent the loading control of immunoblots A and B, respectively. Blots C and D are stained according to as indicated in the Materials and Methods section (*Díaz-Zaragoza et al., 2015a*).

**Table 1** **Identification of spots A, B, C, D of Fig. 2. The sequences are the consensus proteins of two independent analyzes.**

| Spot | Accession | Description |
|------|-----------|-------------|
| A | Q9D7Y7_MOUSE | Peptidase S1 domain-containing protein |
| B | Q9CPN9_MOUSE | Trypsinogen 7 |
| B | E9Q0F0_MOUSE | Keratin 78 |
| C | K2C73_MOUSE | Keratin, type II cytoskeletal 73 |
| C | K1C10_MOUSE | Keratin, type I cytoskeletal 10 |
| D | K2C6B_MOUSE | Keratin, type II cytoskeletal 6B |

mice, integrating epithelial cell proliferation, differentiation, and apoptosis. Sexual maturity is reached at approximately five weeks in mice, although the ductal network continues to grow beyond this point until it reaches its full dimension at around eight weeks (*Richert*

*et al., 2000*). The antigens identified by the IgM of the normal female correspond to this stage of tissue remodeling (5–8 weeks); therefore, the sequenced proteins are part of this enzymatic-structural complex that breast tissue undergoes at the sixth week of age. Both the proteases and keratins found have been reported in the remodeling of the extracellular matrix during the different stages of mammary development and, therefore, in the modulation of the function of mammary cells (*Opsahl et al., 2013*; *Asch & Asch, 1985*; *McNally & Martin, 2011*; *Wiseman et al., 2003*; *Shao et al., 2012*).

It cannot be ruled out that the disappearance of the antigens is due to another reason related to the health of the mouse and not necessarily to cancer in progress; however, although it could be possible, we consider it to be unlikely.

## CONCLUSIONS

In the search for biomarkers that are related to the initial stages of cancer, it is common to try to identify proteins that are modified by the carcinogenesis process or that are expressed de novo. However, the absence of a protein can also be a biomarker. In other words, the cell, in its transformation process, presents an imbalance of its normal protein synthesis machinery and an absence of a protein could be indicative of a neoplasm on the way. In the case of breast tissue, the stages of development are known in detail (both in mice and in humans), so in the transformation process, apparently due to a reorganization of the membrane proteins (see *Da Costa et al., 2000*; *Horníková, Bruštíková & Forstová, 2020*; *Richert et al., 2000*; *Shao et al., 2012*) the proteins that normally should be in the membrane are no longer there and that this could be a way of recognize, identify or predict an impending oncogenesis process.

It is necessary to say that we are not proposing the proteins identified here as biomarkers of the beginning of a transformation process. Much about keratins and breast cancer has already been written. More studies need to be done to make claims to that effect.

### Funding

This work was supported by the Programa de Apoyo a Proyectos de Investigación e Innovación Tecnológica (PAPIIT), the Dirección General de Asuntos del Personal Académico (DGAPA), and the Universidad Nacional Autónoma de México, Ciudad de México, México (grant no. IT200120) to Pedro Ostoa-Saloma. The funders had no role in study design, data collection and analysis, decision to publish, or preparation of the manuscript.

### Grant Disclosures

The following grant information was disclosed by the authors:
Programa de Apoyo a Proyectos de Investigación e Innovación Tecnológica (PAPIIT).
Dirección General de Asuntos del Personal Académico (DGAPA).
Universidad Nacional Autónoma de México, Ciudad de México, México: IT200120.

## Competing Interests

The authors declare there are no competing interests.

## Author Contributions

- Ricardo Hernández Ávila performed the experiments, analyzed the data, prepared figures and/or tables, authored or reviewed drafts of the article, and approved the final draft.
- Mariana Díaz-Zaragoza performed the experiments, analyzed the data, prepared figures and/or tables, authored or reviewed drafts of the article, and approved the final draft.
- Pedro Ostoa-Saloma conceived and designed the experiments, analyzed the data, authored or reviewed drafts of the article, and approved the final draft.

## Animal Ethics

The following information was supplied relating to ethical approvals (*i.e.*, approving body and any reference numbers):

Animal care and experimental practice were conducted at the Unidad de Modelos Biológicos (UMB) in the Instituto de Investigaciones Biomédicas (IIB), Universidad Nacional Autónoma de México.

All experimental procedures in the animals were approved by the Institutional Care and Animal Use Committee (CICUAL), permit number ID 264, adhering to Mexican regulation (NOM-062-ZOO-1999), and in accordance with the recommendations from the National Institute of Health (NIH) of the United States of America.

## Ethics

The following information was supplied relating to ethical approvals (*i.e.*, approving body and any reference numbers):

This study was approved by the Institutional Care and Animal Use Committee (CICUAL) (permit number ID264, (NOM-062-ZOO-1999).

## Data Availability

The raw data is available in the Supplemental Files.

## Supplemental Information

Supplemental information for this article can be found online at http://dx.doi.org/10.7717/peerj.14175#supplemental-information.

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
