# Peer review of "Proteomic analysis of IgM antigens from mammary tissue under pre- and post-cancer conditions using the MMTV-PyVT mouse model"

_PeerJ, doi:10.7717/peerj.14175_

## Round 0.1 · original submission · Minor Revisions

Three reviewers have given their comments and I am happy to see they have provided a few modifications. Please attend to these suggestions and get back to us as soon as possible. An interesting observation by one of the reviewers on "the disappearance of these antigens may be linked to developmental disturbances or any other disease conditions" is very interesting and will be looking forward to an explanation of this intriguing question.

Reviewer 1 ·

Basic reporting

See below

Experimental design

See below

Validity of the findings

See below

Additional comments

The article describes an important observation regarding breast cancer development that certain surface antigens appear and disappear at different time points in the development of tumor. Ths was confirmed by using monoclonal antibodies against specific cell membrane antigens. The authors indicate that the disappearance of these antigens might be used as cancer markers in future. The observations are preliminary in nature and need further experimental validation. The question remains, whether the disappearance of these antigens may be linked to developmental disturbances or any other disease conditions.

·

Basic reporting

No comments

Experimental design

No comments

Validity of the findings

No comments

Additional comments

1. Authors did the smart work on isolating the key observations between the mice models used for the studies.
2. Before going for publication this manuscript needs a co-relation between the metabolite profile as authors have used serum samples for their 2-D and MALDI studies.
3. Un-targeted metabolomics using direct serum samples would be a better fit to co-relate with the existing findings.
4. Metabolomic profile would have become very much conclusive when explained in co-relation to the expression of antigens and other proteins quoted. This would be very much helpful for defining the biomarkers and assuring the outcome of the study.
5. Saying the absence of antigens would be a biomarker needs some more in detail analysis wherein epitome mapping, and cell imaging with histology studies would help.
6. I would say the idea was very much clear but needs furthermore critical analysis for the claims, I strongly feel the manuscript would become more informative to the scientific community with the inclusion of some more aspects to the work for better claims and conclusions.

·

Basic reporting

In addition, the manuscript is clearly written in professional, unambiguous language. If there is a weakness, it is in the experimental and analysis (as I have noted below) which should be improved upon before Acceptance.
The gel pictures are not convincing and need more raw data to be conclusive.

Experimental design

The experimental design is quite simple and lacks proper loading control in 2D Gel pictures. complete 2D profile of the samples stained with silver blue or silver stain need to be provided as Loading control.

Validity of the findings

The data analysis should be improved in the following ways:
Table 1 has A and B in the first and 2nd row as Peptidase S1 domain whereas 3rd Row mentioned as B is Keratin 78, Kindly verify, as in the gel picture, the 2 spots are distinctly separate and non-overlapping.
Also, Kindly provide the complete MS profile and data set.

The Immunoblot pictures does not look convincing, also the loading control is missing. Also, is there any biomarker, which can be used as loading control?
Will the Post translation modification affect the antigenicity of these proteins against IgM? Can you please provide validated data for the same.
As it is evident from the gel picture , that the transgenic mice gel picture, over all the protein expression is less.

Where the spots above A and B analyzed, as even they show significantly low expression of protein in the Gel B, if yes, kindly provide the data and if not, kindly explain why?

Conclusion is as per the the research questions, but still need further verification. The protein identified, need more description and functional aspects need to be discussed.

Additional comments

No comments

---

## Round 0.2 · accepted · Accept

This paper is recommended for publication

·

Basic reporting

No comments

Experimental design

No comments

Validity of the findings

No comments

Additional comments

No further comments

·

Basic reporting

Clear and unambiguous, professional English used throughout.
Self-contained with relevant results and hypothesis

Experimental design

Research question well defined, relevant & meaningful. It is stated how research fills an identified knowledge gap

Validity of the findings

All underlying data have been provided; they are robust, statistically sound, & controlled.